# Development of a True-Biaxial Split Hopkinson Pressure Bar Device and Its Application

**DOI:** 10.3390/ma14237298

**Published:** 2021-11-29

**Authors:** Shumeng Pang, Weijun Tao, Yingjing Liang, Shi Huan, Yijie Liu, Jiangping Chen

**Affiliations:** 1School of Environment and Civil Engineering, Dongguan University of Technology, Dongguan 523808, China; mengps0812@hotmail.com; 2School of Civil Engineering, Guangzhou University, Guangzhou 510006, China; liangyingjing@126.com (Y.L.); huanshi@gzhu.edu.cn (S.H.); liuyijie1987@outlook.com (Y.L.); cjp81@126.com (J.C.)

**Keywords:** true-biaxial split Hopkinson pressure bar, dynamic mechanical properties, wedge-shaped dual-wave bar, shear stress wave, axial stress wave

## Abstract

Although highly desirable, the experimental technology of the dynamic mechanical properties of materials under multiaxial impact loading is rarely explored. In this study, a true-biaxial split Hopkinson pressure bar device is developed to achieve the biaxial synchronous impact loading of a specimen. A symmetrical wedge-shaped, dual-wave bar is designed to decompose a single stress wave into two independent and symmetric stress waves that eventually form an orthogonal system and load the specimen synchronously. Furthermore, a combination of ground gaskets and lubricant is employed to eliminate the shear stress wave and separate the coupling of the shear and axial stress waves propagating in bars. Some confirmatory and applied tests are carried out, and the results show not only the feasibility of this modified device but also the dynamic mechanical characteristics of specimens under biaxial impact loading. This novel technique is readily implementable and also has good application potential in material mechanics testing.

## 1. Introduction

In practical engineering, structural materials are generally in a complex three-dimensional (3D) stress state. When a structure is subjected to blast or impact loads, its materials are in an even more complex 3D state because of the strain rate effect, and it is difficult to simulate and analyze the stress state accurately with the material constitutive relationship obtained using a traditional impact-loading experimental device. This is because (i) it is difficult to carry out 3D impact loading tests using existing experimental devices, and (ii) the two-dimensional and 3D models that are used are ultimately based on existing strength theories from one-dimensional (1D) impact loading tests. Hence, a question that requires research is whether the strength model, strain rate effect, and failure criterion of a material under multi-dimensional impact loading are consistent with the theories developed under 1D impact loading. Therefore, to study the dynamic strength of materials under multi-dimensional impact loading, we must develop a multi-dimensional impact-loading experimental device.

Since Kolsky [1] finalized the split Hopkinson pressure bar (SHPB), it has become a commonly used experimental device for studying the dynamic mechanical properties of different materials under different strain rates, such as concrete and rock [2,3,4,5], metals [6,7,8], and composite materials [9,10,11]. It has also been modified into various experimental devices for testing the dynamic mechanical properties of materials under complex stress loading.

Christenson et al. [12] improved the traditional SHPB device by adding a pressure vessel so that all the principal stresses and strains of rocks could be recorded under confining pressure. To investigate the dynamic mechanical properties of concrete under multiaxial loading, Gary et al. [13] located the specimen in a cylindrical quasi-static pressure cell and developed a specific device that could produce dynamic compression under different lateral confining pressures, allowing radial inertia and lateral pressure effects to be evaluated independently; however, the casing strength and sealing technology meant that the device could not achieve high confining pressure. Based on the passive confining technology applied to an SHPB as proposed by Gong and Malvern [14], Chen and Ravichandran [15] used steel and aluminum jackets to improve the lateral restraint strength of cylindrical specimens; however, the required high machining precision and the friction effect between the specimen and the jackets meant that the tests were not as repeatable or reliable as desired. Shi et al. [16] applied oil lipid antirust grease evenly on the outer surface of the specimen and used the oil film as the coupling medium to transfer the confining pressure in the gap between the specimen and the jackets; this method reduced the friction between the specimen and the jackets and met the required machining precision. Forquin [17] filled the gap between a concrete specimen and a metallic ring with an epoxy resin to reach a higher confining pressure, as well as gluing transverse gauges on the lateral surface of the metallic ring to record the confining pressure. This passive confining-pressure technique has been applied to the dynamic mechanical properties of concrete materials [18,19,20,21]. Chen et al. [22] discussed the confining-pressure testing technique for soft materials under triaxial loading. Nemat-Nasser and Rome [23] combined active and passive confining pressures and designed a triaxial confining-pressure Hopkinson experimental device that could generate a higher lateral confining pressure on the specimen. Shi et al. [24] improved the SHPB device with a special active hydraulic confining-pressure installation, and loaded different lateral confining pressures on blends. Li et al. [25,26,27] improved the traditional Hopkinson pressure bar and applied both axial static pressure and lateral confining pressure on the specimen simultaneously. The active confining-pressure technique has also been used to research the dynamic mechanical properties of rocks [28,29,30], glass beads [31], and artificial frozen silty clay [32,33,34]. Albertini et al. [35,36] designed a 3D static and dynamic experimental device that could pre-load axial pressure on the specimen in three dimensions under 1D impact loading. This device could also measure the lateral stress wave signals produced by the incident stress wave loading on the specimen [37,38]; compared with the results from uniaxial, biaxial, and triaxial tests, the characteristics of elastic modulus, compressive strength, and failure modes of materials are related to the different constrain conditions of confining pressures [39,40].

In the aforementioned experimental devices, the key technology is that the specimen is preloaded with a static confining pressure either laterally or axially and is then loaded with an axial incident stress wave. This is a typical coupled static–dynamic experimental technology and is still a 1D impact test. However, when an engineering structure is subjected to dynamic loads, its materials are in a complex 3D dynamic stress state, and the dynamic mechanical properties of materials should also be 3D dynamics. Therefore, research is required into experimental technology for loading specimens three-dimensionally.

Hummeltenberg et al. [41] designed a biaxial SHPB experimental device that comprised two gas guns, two striker bars, two incident bars, and two transmission bars, with a cube specimen placed between the two incident bars and the two transmission bars; however, the errors inherent in the gas driving system made it difficult to produce two incident stress waves that would load the specimen simultaneously. Huan et al. designed a triaxial SHPB experimental device that used a striker bar to impact three incident bars simultaneously and produced three incident stress waves propagating synchronously in the incident bars [42,43]; these three incident stress waves were passed to the cube specimen through three steering heads placed between the cube specimen and the three incident bars. That experimental device was capable of loading the specimen synchronously with three stress waves, however the steering heads produced a shear stress wave that was coupled to the axial stress wave in the bars, thereby complicating the data processing and experimental analysis. Li et al. [44,45,46,47] developed a 2D/3D SHPB experimental device based on an electro-magnetic riveting method, which used electromagnetic energy conversion technology to generate a stress wave pulse, which requires higher circuit control accuracy and a more accurate electromagnetic riveting device.

Therefore, the experiences with the aforementioned biaxial and triaxial experimental devices indicate that developing a multi-dimensional impact-loading experimental device requires solving two experimental technology problems, namely (i) how to synchronize the propagation of the incident stress waves and (ii) how to eliminate the propagation coupling between shear and axial stress waves in the bars. In the present study, a wedge-shaped dual-wave bar (DWB) was designed, and a biaxial SHPB experimental device was developed that solves the problem of propagation coupling between shear and axial stress waves, realizing the simultaneous loading of two incident stress waves on a cube specimen. Furthermore, the device was used in an experiment to determine the dynamic mechanical properties of beech wood.

## 2. Design of Dual-Wave Bar and Its Experimental Analysis

### 2.1. Design of Wedge-Shaped, Dual-Wave Bar

According to the analysis in Section 1, when the striker bar strikes two or three incident bars, the stress waves that are produced are the same and meet the synchronization requirement. However, the data processing is complicated by the shear stress wave produced in the steering heads from the incident bar to the cube specimen. Zhao and Lu et al. [48,49] proposed a compression-shear experimental device with a wedge-shaped incident bar and two transmission bars, which could not only apply compression-shear load to the specimen, but also generate two transmission stress waves that propagate synchronously, the design of which can provide a good reference for the research of this paper. In this paper, the method including a striker bar strikes two incident bars or three incident bars is referenced, and a DWB is designed and applied into the development of biaxial SHPB. As shown in Figure 1, one end of the DWB is designed as a symmetrical wedge-shaped section; each of the two symmetrical sections is connected to a transmission bar, and the structure is symmetrical. When the striker strikes the DWB at some impact speed, an incident stress wave is produced that then propagates in the DWB. The impedance mismatch between the DWB and the transmission bars means that the incident stress wave is reflected and transmitted at the interface between the wedge-shaped section and the transmission bars. The incident stress wave is reflected in part into the DWB and transmitted in part through the transmission bars. Based on the physical mechanism for the structural symmetry, the two stress waves propagating in the transmission bars are, in theory, the same and thus synchronous. This method is referred to as the wave decomposition technique.

### 2.2. Synchronicity of Stress Wave Propagation

As shown in Figure 2, each transmission bar had a set of strain gauge (TP-3.8-120) glued symmetrically to it at the same distance from the wedge-shaped section of the DWB; the distance of L is 712 mm. Moreover, the distance between the strain gauge glued on the DWB and the wedge-shape section is 690 mm. The material parameters and the bar size are given in Table 1 and the striker bar 1 is adopted in this test. When the striker bar struck the DWB at a certain speed, the resulting incident stress wave was transmitted through the wedge-shaped section to the two transmission bars. Figure 3 depicts the stress wave signals measured by the strain gauges, in which the two transmission stress waves coincided, indicating that the transmission stress waves derived from the DWB decomposition are identical and meet the requirement of synchronous propagation.

### 2.3. Analysis for Propagation of Stress Wave

The incident stress wave is reflected and transmitted at the interfaces between the wedge-shaped section and the transmission bars when it arrives at the wedge-shaped section. The difference between the propagation direction of the incident stress wave and the normal direction of the wedge-shaped section generates an axial stress wave in the axial direction of the transmission bar and a shear stress wave perpendicular to that axial direction. Some basic tests were conducted to evaluate how shear stress waves affect the stress wave signals measured by strain gauges. As shown in Figure 4, two semiconductor strain gauges were glued symmetrically on the horizontal surface of each side of the transmission bars. The single-crystal strip direction of each semiconductor strain gauge was parallel to the axial direction of the transmission bar. Three sets of strain gauges were attached to the transmission bar with distances of 153 mm, 266.5 mm, and 386.5 mm from the wedge-shaped portion, respectively. The material parameters and the bar size are given in Table 1, and the striker bar 2 is used in this test.

When there is no lubricant at the interface between the wedge-shaped section of the DWB and a transmission bar, the striker bar strikes the DWB at a certain speed and the resulting incident stress wave is transmitted through the wedge-shaped section to each transmission bar. The stress wave signals measured by the strain gauges are shown in Figure 5. At the beginning section of the rising edge, the two stress wave signals detected by the two glued strain gauges coincide symmetrically, and the stress wave component of the coincidence part is an axial stress wave signal propagating independently. Then, the non-coincidence part of the two curves produces different trends at some point in time. This phenomenon arises because the shear stress wave propagates to the positions of the strain gauges and affects the stress wave signals that they measure. Each stress wave signal couples the shear stress wave and the axial stress wave. As the propagation distance increases, more parts of the stress wave signal coincide with each other, which can be understood from the fact that, when the propagation distance is long enough, the axial stress wave and the shear stress wave are decomposed and propagate independently. The stress wave signals measured by the three groups of strain gauges were linearly superimposed and averaged, and the resulting curves are shown in Figure 6. The three curves have similar trends and amplitudes, indicating that the two symmetrically bonded strain gauges can filter out the shear stress wave’s influence. However, the shear stress wave persists in the bar and might load the specimen, compromising the test results’ accuracy and validity.

The striker bar impacts the DWB at a certain speed when there is lubricant at the interface between the wedge-shaped portion of the DWB and a transmission bar, and the consequent incident stress wave is transmitted via the wedge-shaped section to each transmission bar. The stress wave signals measured by the strain gauges are shown in Figure 7. Compared with Figure 5, the curves in Figure 7 show the shear stress wave having less of an effect on the signals measured by the strain gauges. The lubricant reduces (i) the interaction between the wedge-shape section and the transmission bars and (ii) the generation and transmission of the shear stress wave. However, the shear stress wave remains in the bar and cannot be eliminated.

More studies on how to eradicate the shear stress wave are needed to reduce the shear stress and its influence on the test data. The shear stress waves significantly weakened as they propagated through multiple layers of lubricant at the interface. A circular gasket with the dimensions of Ø25 × 0.1 mm was designed, and three of them were placed between the wedge-shaped potion and a transmission bar, with their interfaces equally coated with lubricant, as displayed in Figure 8. When the striker bar strikes the DWB at a certain speed, the resulting incident stress wave is transmitted through the wedge-shaped section and the round gaskets to each transmission bar. The stress wave signals measured by the strain gauges are shown in Figure 9. The two stress wave signals obtained by the two symmetrically glued strain gauges coincide, implying that the shear stress wave can be eliminated utilizing the multilayer round gaskets and lubrication approach.

## 3. Biaxial SHPB Experimental Device

A true-biaxial SHPB experimental device was developed based on the design of the wedge-shaped DWB and the analysis of the stress wave propagation in Figure 10. This experimental device comprises a launch system, a striker bar, three DWBs, two balance bars, two incident bars, two transmission bars, and a measurement system. The material parameters of the pressure bars are summarized in Table 2.

During testing, the launch system drives the striker bar to impact DWB A, generating a single stress wave that propagates through the structure. When this stress wave reaches the wedge-shaped portion, a compression stress wave propagated synchronously to DWB B1 and DWB B2, which is the first stress-wave decomposition at F. When the compression wave was propagating, the second stress wave decomposition was performed at F1 and F2, respectively, and then two incident stress waves were formed, which propagated in the incident bar D1 and D2, respectively, and load the cube specimen, respectively. The two stress waves are reflected and transmitted because of the impedance mismatch between the cube specimen and the pressure bars, resulting in two reflected stress waves and two transmitted stress waves. The measurement system includes some strain gauges, a high dynamic strain indicator (KD6009), a DPO2000 oscilloscope, and a laser velocimeter. The strain gauges on the two incident bars and the two transmission bars monitor the stress wave pulses propagating through the pressure bars, which are subsequently transmitted to the DPO2000 oscilloscope via the high dynamic strain indicator. The distances between the four sets of strain gauges and the specimen are same and are equal to 30 cm. Moreover, the impact velocity of the striker bar is measured by the laser velocimeter.

## 4. Theory Study of Biaxial Impact Loading

Figure 11 depicts the two incident stress waves propagating in the two incident bars and, subsequently, the loading cube specimen. The impedance mismatch between the bars and the cube specimen produces a reflection and transmission of the stress wave at the interfaces between the specimen and the pressure bars. A portion of the incident stress waves is reflected into the incident bars, forming the reflected stress waves, while the other part is transmitted through the transmission bars, forming the transmitted stress waves. The stress wave signals were measured by strain gauges glued at the middle positions of the bars. In the *x*-direction are the incident wave signal $\varepsilon_{Ix}$, the reflected wave signal $\varepsilon_{Rx}$, and the transmitted wave signal $\varepsilon_{Tx}$, and in the *y*-direction are the incident wave signal $\varepsilon_{Iy}$, the reflected wave signal $\varepsilon_{Ry}$, and the transmitted wave signal $\varepsilon_{Ty}$. Assuming that the stress wave propagation in the *x* and *y* directions satisfies the 1D stress-wave theory [39,50], the strain rates ${\dot{\varepsilon }}_{x}\left(t\right)$ and ${\dot{\varepsilon }}_{y}\left(t\right)$, the strain $\varepsilon_{x}\left(t\right)$ and $\varepsilon_{y}\left(t\right)$, and the stress $\sigma_{x}\left(t\right)$ and $\sigma_{y}\left(t\right)$ of the specimen can be deduced in the *x* and *y* directions, respectively. The equations for doing so are:
(1)$${\dot{\varepsilon }}_{x}\left(t\right)=\frac{C_{0}}{l}\left[\varepsilon_{Ix}\left(t\right)-\varepsilon_{Rx}\left(t\right)-\varepsilon_{Tx}\left(t\right)\right]$$
(2)$$\varepsilon_{x}\left(t\right)=\frac{C_{0}}{l}\int_{0}^{t}\left[\varepsilon_{Ix}\left(t\right)-\varepsilon_{Rx}\left(t\right)-\varepsilon_{Tx}\left(t\right)\right]dt$$
(3)$$\sigma_{x}\left(t\right)=\frac{A_{0}}{2A_{s}}E_{0}\left[\varepsilon_{Ix}\left(t\right)+\varepsilon_{Rx}\left(t\right)+\varepsilon_{Tx}\left(t\right)\right]$$
in the *x* direction and
(4)$${\dot{\varepsilon }}_{y}\left(t\right)=\frac{C_{0}}{l}\left[\varepsilon_{Iy}\left(t\right)-\varepsilon_{Ry}\left(t\right)-\varepsilon_{Ty}\left(t\right)\right]$$
(5)$$\varepsilon_{y}\left(t\right)=\frac{C_{0}}{l}\int_{0}^{t}\left[\varepsilon_{Iy}\left(t\right)-\varepsilon_{Ry}\left(t\right)-\varepsilon_{Ty}\left(t\right)\right]dt$$
(6)$$\sigma_{y}\left(t\right)=\frac{A_{0}}{2A_{s}}E_{0}\left[\varepsilon_{Iy}\left(t\right)+\varepsilon_{Ry}\left(t\right)+\varepsilon_{Ty}\left(t\right)\right]$$
in the *y*-direction, where *C*_0_ is the wave speed in a bar and *l* and *A*_s_ are the specimen length and area, respectively.

To better describe the dynamic mechanical properties of the cube specimen under biaxial impact loading, the effective strain and the effective stress are employed, namely
(7)$$\bar{\varepsilon }=\sqrt{\frac{4}{3}J_{2}^{\prime }}$$
(8)$$\bar{\sigma }=\sqrt{3J_{2}}$$
where $J_{2}^{\prime }$ and $J_{2}$ are the second invariants of the deviatoric strain tensor and the deviatoric stress tensor, respectively, namely
(9)$$J_{2}^{\prime }=\frac{1}{6}\left[{\left(\varepsilon_{x}\left(t\right)-\varepsilon_{y}\left(t\right)\right)}^{2}+{\left(\varepsilon_{y}\left(t\right)-\varepsilon_{z}\left(t\right)\right)}^{2}+{\left(\varepsilon_{z}\left(t\right)-\varepsilon_{x}\left(t\right)\right)}^{2}\right]$$
(10)$$J_{2}=\frac{1}{6}\left[{\left(\sigma_{x}\left(t\right)-\sigma_{y}\left(t\right)\right)}^{2}+{\left(\sigma_{y}\left(t\right)-\sigma_{z}\left(t\right)\right)}^{2}+{\left(\sigma_{z}\left(t\right)-\sigma_{x}\left(t\right)\right)}^{2}\right]$$

Finally, the effective strain rate of the cube specimen under biaxial impact loading is
(11)$$\bar{\dot{\varepsilon }}=\frac{1}{3}\sqrt{2\left[{\left({\dot{\varepsilon }}_{x}-{\dot{\varepsilon }}_{y}\right)}^{2}+{\left({\dot{\varepsilon }}_{y}-{\dot{\varepsilon }}_{z}\right)}^{2}+{\left({\dot{\varepsilon }}_{z}-{\dot{\varepsilon }}_{x}\right)}^{2}\right]}$$

## 5. Experimental Results and Analysis

### 5.1. Effectiveness Analysis

The selected test material was natural beech wood, which is a macroscopically orthotropic material. The wooden specimens were shaped into cubes and machined. The cube specimen has a side length of 12 mm. As shown in Figure 12, the specimen was subjected to impact loads in both the radial and tangential directions. The striker bar struck the DWB at a certain speed and the resulting incident stress wave propagated to the two incident bars and formed two incident stress waves after two wave decompositions. The specimen was loaded by these two incident stress waves, which resulted in the transmission and reflection of the stress waves. The stress wave signals recorded by strain gauges attached to the two incident bars and the two transmission bars are shown in Figure 13.

Figure 13 shows that the two incident stress waves were triggered at the same instant and had the same magnitude, which satisfies the requirement of the synchronicity of stress wave propagation. The two transmitted stress waves had similar trends with different amplitudes, and the same was true of the two reflected stress waves. The amplitude of the transmitted stress wave in the radial direction was higher than that in the tangential direction, whereas the amplitude of the reflected stress wave in the radial direction was smaller than that in the tangential direction. These characteristics are due to the different physical properties of beech wood in the radial and tangential directions.

Figure 13 also shows the wave profiles of the incident, reflected, and transmitted stress waves in the radial and tangential directions of the wooden specimen. According to the three-wave method, stress equilibrium in each direction is achieved to some extent, which illustrates the validity of the experimental data.

### 5.2. Application of Biaxial SHPB

As shown in Figure 12, the beech wood specimen was subjected to impact loads simultaneously in the radial and tangential directions. Performed a series of tests with different impact speeds, the stress wave signals were measured by the stain gauges attached to the two incident bars and the transmission bars. The calculated results using the theoretical formulas in Section 4 are shown in Figure 14 and Figure 15.

The stress–strain curves of beech wood specimens in the radial and tangential directions under various strain rates are shown in Figure 14. In each test, the strain rate and maximum strain of the beech wood in the tangential direction were greater than those in the radial direction, while the ultimate strength of the beech wood in the tangential direction was lower than that in the radial direction, which shows that the mechanical properties of beech wood in the radial and tangential direction are different, which is consistent with the transverse anisotropy of beech wood. In addition, the ultimate strength of the beech wood in each direction increased with the strain rate. Figure 15 shows the effective stress–strain curves of the beech wood specimens in the radial and tangential directions at various effective strain rates. The effective ultimate strength and the effective maximum strain increased as the effective strain rate increased, indicating that the dynamic mechanical properties of beech wood are rate-dependent.

## 6. Conclusions

A true-biaxial SHPB experimental device was developed in the present study. Based on the wave decomposition technique, the wedge-shaped DWB could decompose a single stress wave into two stress waves propagating synchronously in two pressure bars. The combination of several round gaskets and lubricant between the wedge-shaped section and the pressure bars prevented further transmission of the shear stress wave to the pressure bars, thereby eliminating the shear stress wave and separating the coupling of the shear and axial stress waves propagating in the pressure bars.

The true-biaxial SHPB device could load a cube specimen synchronously with two orthogonal stress waves, and the incident, reflected, and transmitted stress wave signals were measured completely by the strain gauges. The validity of the experimental data showed that the experimental device was feasible to measure the dynamic mechanics of materials under biaxial loadings. In the same test, a beech wood specimen was subjected to biaxial impact loading in the radial and tangential directions synchronously, giving rise to different strain rates and different stress–strain curves, which could reflect the different dynamic mechanical properties of beech wood in two directions simultaneously. The curves of effective strain versus effective stress were also given, revealing the strain-rate dependence.

This modified device opens opportunities for researching the dynamic mechanical properties of solid materials (including metals, rocks, metamaterials, composite materials, etc.) under biaxial impact loading. It also provides a design reference for developing triaxial or multiaxial impact loading devices.

## Figures and Tables

**Figure 1 materials-14-07298-f001:**
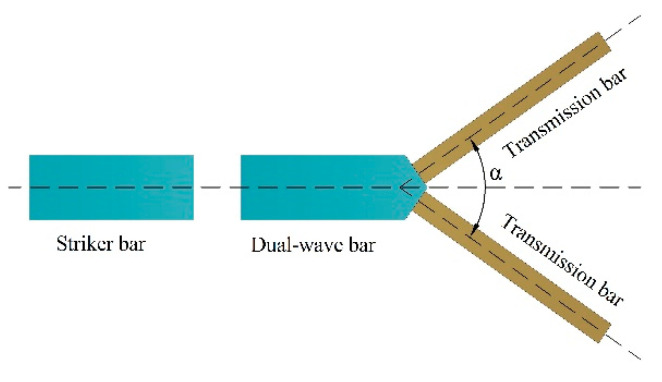
Schematic of wedge-shaped dual-wave bar.

**Figure 2 materials-14-07298-f002:**
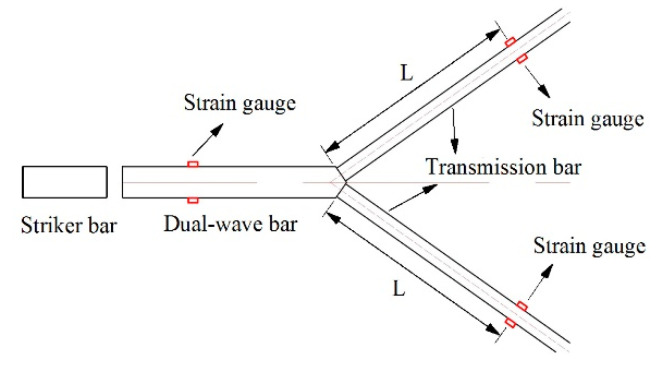
Positions of strain gauges on bars.

**Figure 3 materials-14-07298-f003:**
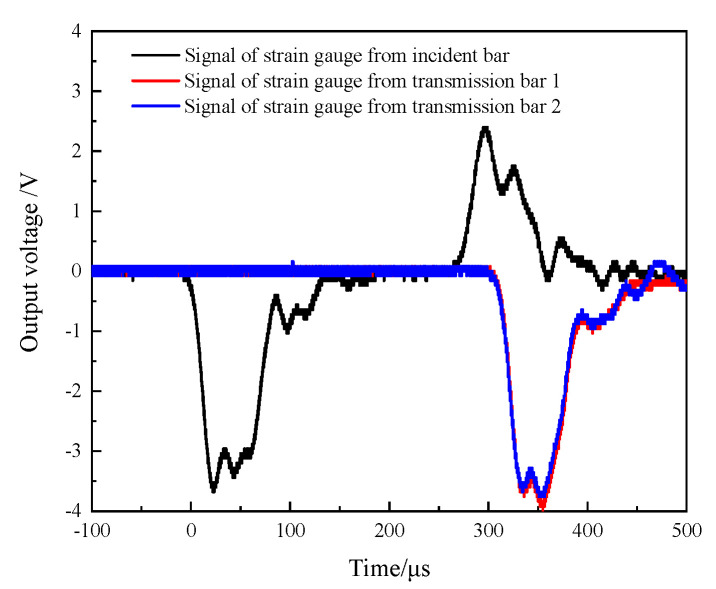
Stress wave signals.

**Figure 4 materials-14-07298-f004:**
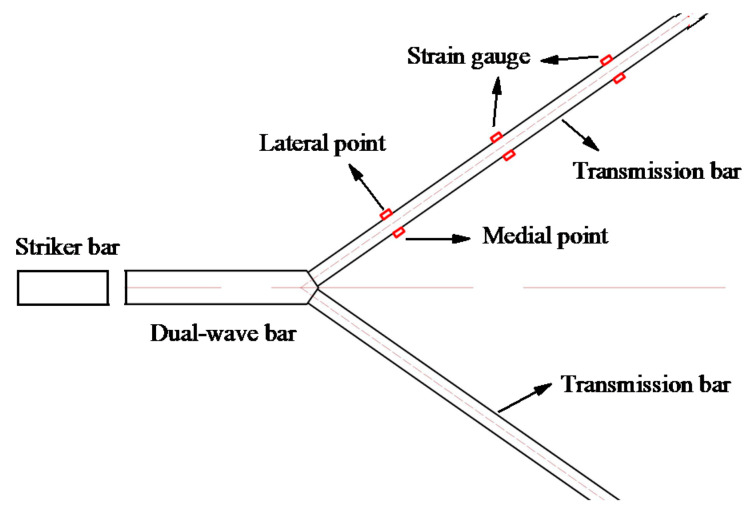
Positions of strain gauges on the transmission bar.

**Figure 5 materials-14-07298-f005:**
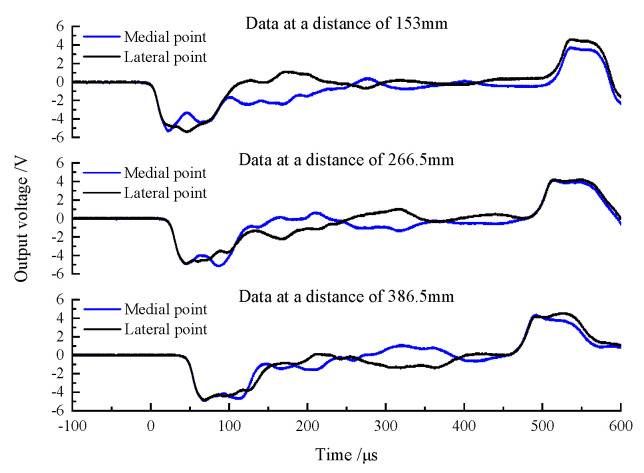
Wave pulse signals of lateral point and medial point at different distances (without lubricant).

**Figure 6 materials-14-07298-f006:**
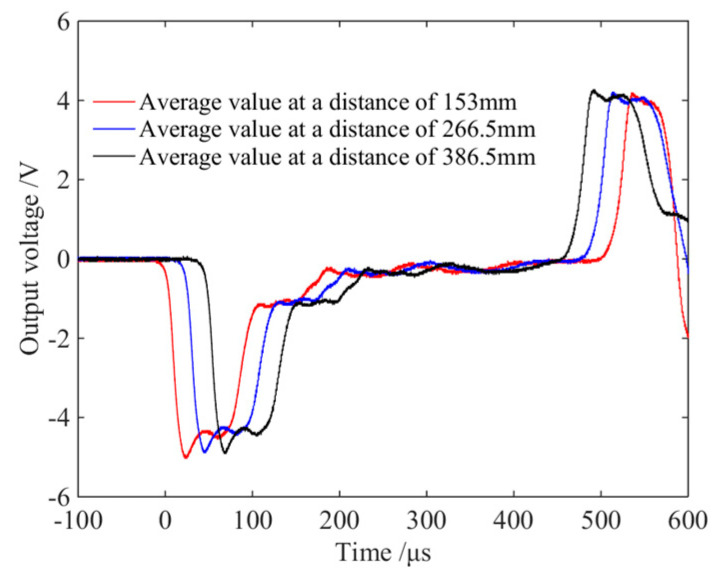
Average values of wave pulse signals of lateral point and medial point at different distances.

**Figure 7 materials-14-07298-f007:**
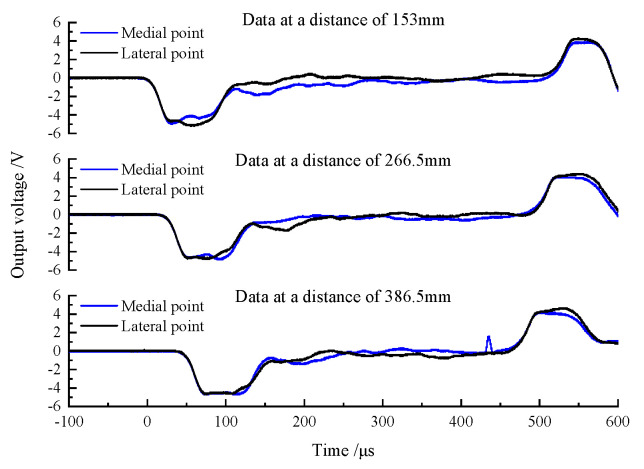
Wave pulse signals of lateral point and medial point at different distances (with lubricant).

**Figure 8 materials-14-07298-f008:**
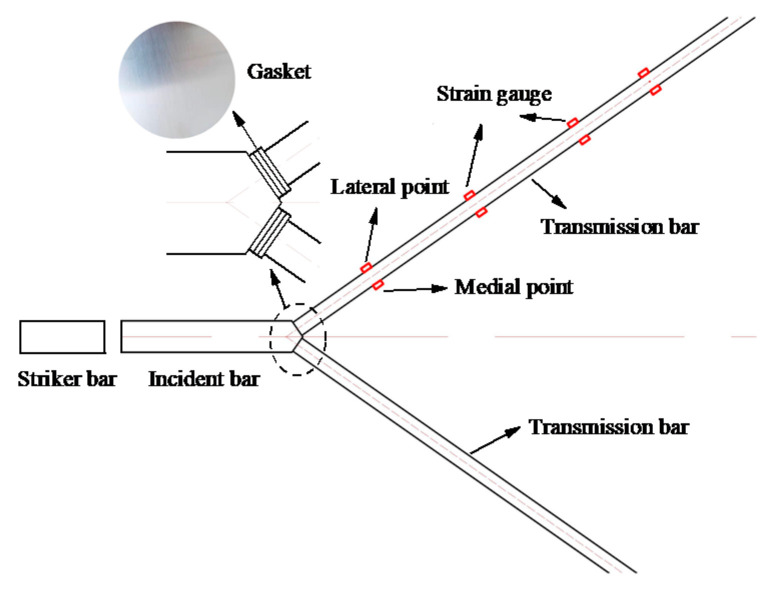
Diagram of gaskets and their placement.

**Figure 9 materials-14-07298-f009:**
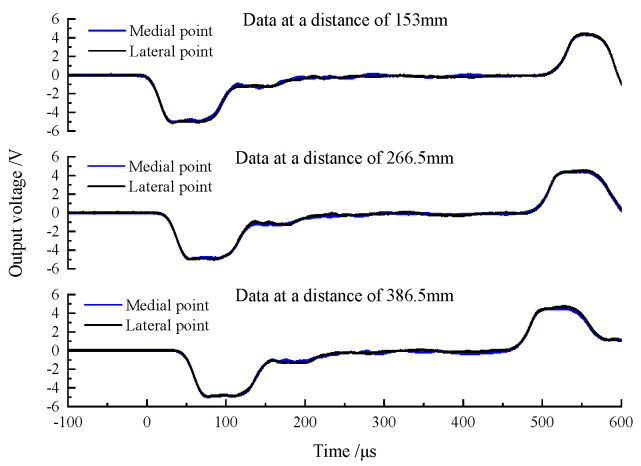
Wave pulse signals of lateral point and medial point at different distances (with gaskets).

**Figure 10 materials-14-07298-f010:**
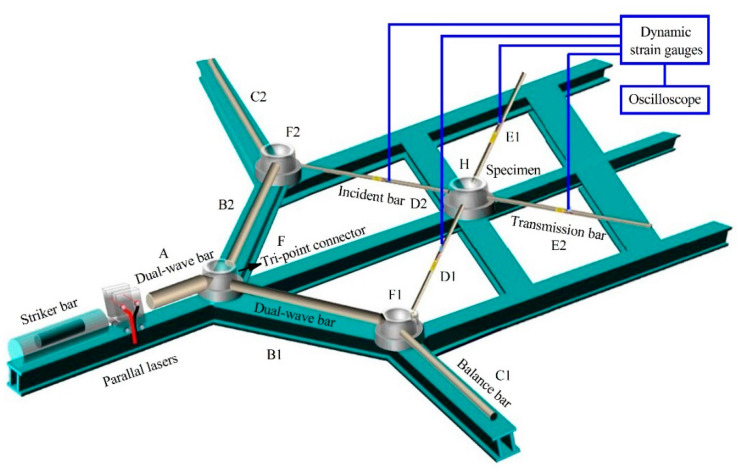
Schematic and photo of biaxial SHPB device.

**Figure 11 materials-14-07298-f011:**
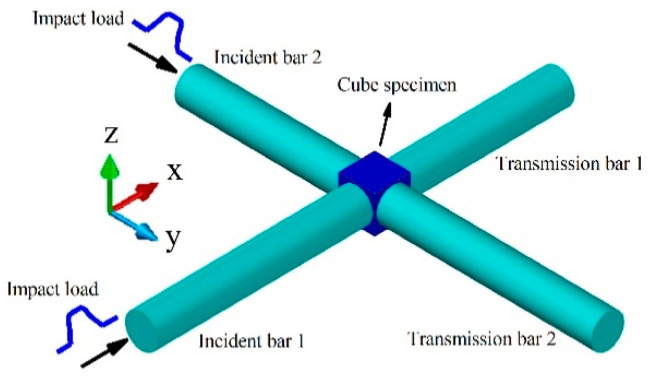
Schematic of biaxial impact experiment.

**Figure 12 materials-14-07298-f012:**
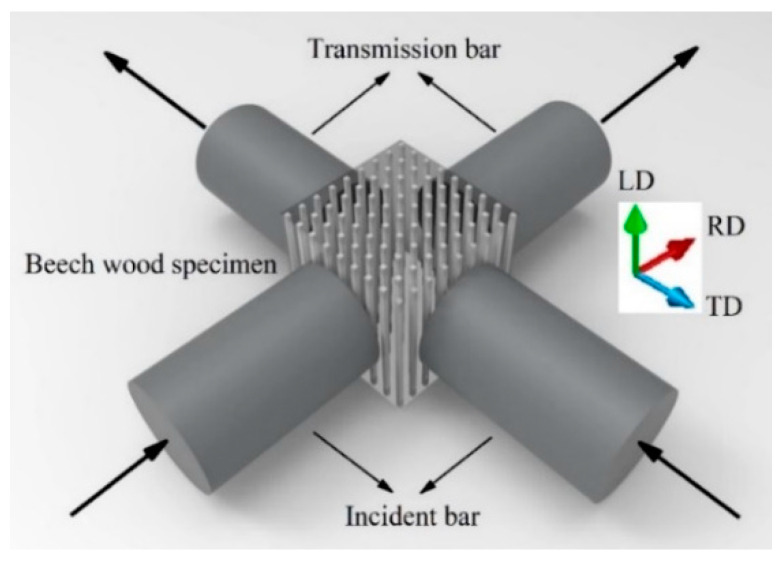
Schematic of placement of beech wood specimen.

**Figure 13 materials-14-07298-f013:**
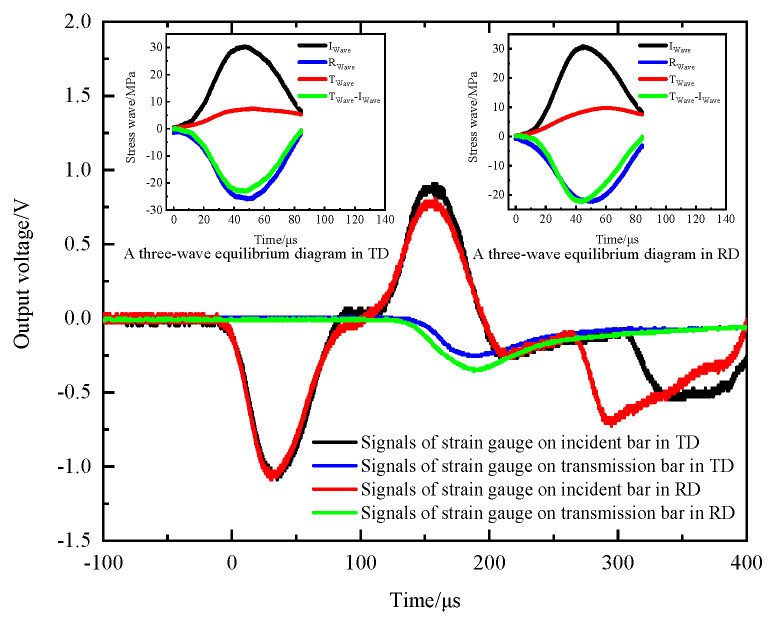
Original signals of stress waves.

**Figure 14 materials-14-07298-f014:**
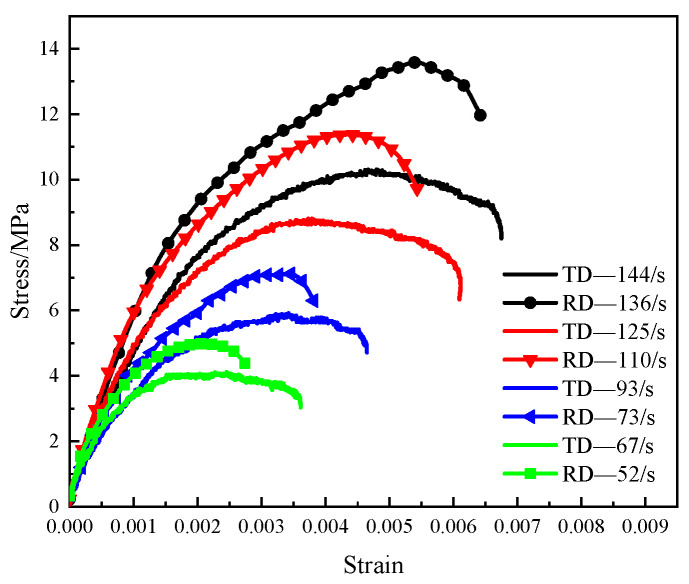
Stress–strain curves of specimens under biaxial impact loading.

**Figure 15 materials-14-07298-f015:**
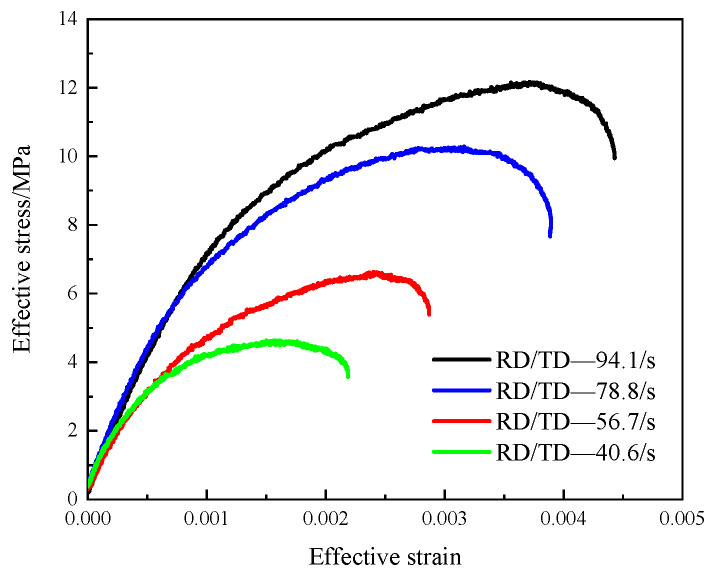
Effective stress–strain curves of specimens under biaxial impact loading.

**Table 1 materials-14-07298-t001:** Material parameters and bar size.

	Material	Density [g/cm^3^]	Young’s Modulus [GPa]	Poisson’s Ratio	Diameter × Length [mm × mm]	Angle [°]
Striker bar 1	SUS304	7.93	193	0.3	Ø50 × 150	-
Striker bar 2	SUS304	7.93	193	0.3	Ø50 × 200	-
DWB	SUS304	7.93	193	0.3	Ø40 × 1514.14	70
Transmission bar	SUS304	7.93	193	0.3	Ø20 × 1499.61	-

**Table 2 materials-14-07298-t002:** Material parameters and bar size.

	Material	Density [g/cm^3^]	Young’s Modulus [GPa]	Poisson’s Ratio	Diameter × Length [mm × mm]	Angle [°]
Striker bar	SUS304	7.93	193	0.3	Ø50 × 200	-
DWB A	SUS304	7.93	193	0.3	Ø40 × 1514.14	70
DWB B1/B2	SUS304	7.93	193	0.3	Ø20 × 1499.61	160
Balance bar C1/C2	SUS304	7.93	193	0.3	Ø10 × 1210	-
Incident bar D1/D2	SUS304	7.93	193	0.3	Ø10 × 1210	-
Transmission bar E1/E2	SUS304	7.93	193	0.3	Ø10 × 1210	-

## Data Availability

The data that support the finding of this study are available from the corresponding author upon reasonable request.

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
