# Peer review of "Development of a True-Biaxial Split Hopkinson Pressure Bar Device and Its Application"

_materials, 2021, doi:10.3390/ma14237298_

Round 1

Reviewer 1 Report

The authors should improve the language throughout the manuscript.

Motivation for the research is unknown? Various researches have been done on a similar topic. What is/are novelty?

Authors should consider to rename the different sections.

Source(s) should be provided for section 4.

Section 5 should be more descriptive. Authors should not only report the results. Reasons for the observed results should be discussed. The authors have not mentioned any scientific reasons for the observed reults.

Section 6 should be renamed as CONCLUSIONS.

Conclusions should be written pointwise highlighting the major findings of the research. 

Author Response

Thank you very much for your positive comments on this manuscript, and we have also revised the manuscript according to your suggestions. Details can be seen as follows:

Question 1:

“The authors should improve the language throughout the manuscript.”

Response 1:

Thank you for this comment. According to your advice, English expression has been carefully improved throughout the manuscript. Details can be seen in this revised manuscript.

Question 2:

“Motivation for the research is unknown? Various researches have been done on a similar topic. What is/are novelty?”

Response 2:

we appreciate this comment. This motivation for this research is that, compared with the traditional experimental device, this research developed a new tre-biaxial experimental device based on a physical wave decomposition method, realizing that the incident stress waves in two directions could load the specimen synchronously, which meets the requirements of biaxial impact loading test.

So far, there have been four researches similar to this research, the first research is that Albertini et al develop a 3D static and dynamic experimental device based on an improvement of the passive confining-pressure technique and active confining-pressure technique, although this device could measure the lateral stress wave signals produced by the 1D incident stress wave loading, the specimen is subjected to impact loads in one direction only, not two. This is a typical coupled static–dynamic experimental technology and is still a 1D impact test.

The second research is that Hummeltenberg et al designed a biaxial SHPB experimental device that comprised two gas guns, two striker bars, two incident bars, and two transmission bars, with a cube specimen placed between the two incident bars and the two transmission bars. However, the errors inherent in the gas driving system made it difficult to produce two incident stress waves that would load the specimen simultaneously.

The third research is that Huan et al. designed a triaxial SHPB experimental device that used a striker bar to impact three incident bars simultaneously and produced three incident stress waves propagating synchronously in the incident bars. That experimental device was capable of loading the specimen synchronously with three stress waves, but the steering heads produced a shear stress wave that was coupled to the axial stress wave in the bars, thereby complicating the data processing and experimental analysis.

The fourth research is that a 2D and 3D SHPB experimental device based on electro-magnetic riveting method was proposed by Li et al, the electro-magnetic riveting technology uses electromagnetic energy conversion technology to generate stress wave pulse, which requires higher circuit control accuracy and more accurate electromagnetic riveting device.

Compared with these researches above, this research developed a new true-biaxial experimental device based on a physical wave decomposition method, which not only realizes that the incident stress waves in two directions could load the specimen synchronously, but also eliminates the influence of shear stress wave on test data, which meets the requirements of biaxial impact loading test.

Question 3:

“Authors should consider to rename the different sections.”

Response 3:

Thanks for this comment. According to your advice, we have renamed some of the sections.

Question 4:

“Source(s) should be provided for section 4.”

Response 4:

Thanks for this comment. According to your advice, we have cited two relevant articles as references. Details can be seen in this revised manuscript (L15-L17 in section 4)

Question 5:

“Section 5 should be more descriptive. Authors should not only report the results. Reasons for the observed results should be discussed. The authors have not mentioned any scientific reasons for the observed results.”

Response 5:

Thanks for this comment. According to your advice, we have modified some of the expressions and gave the reason why the strain rate and stress-strain curves of beech wood  were different in two directions under biaxial impact loading. Details can be seen in this revised manuscript (L45-L68 and L80-L92 in section 5).

Question 6:

“Section 6 should be renamed as CONCLUSIONS.”

Response 6:

Thanks for this comment. According to your advice, we have renamed the section 6 as CONCLUSIONS.

Question 7:

“Conclusions should be written pointwise highlighting the major findings of the research. ”

Response 7:

Thanks for this comment. According to your advice, we have modified some of the expressions to highlight the main findings of the study. Details can be seen in this revised manuscript (L97-L118).

Reviewer 2 Report

The paper describes a novel biaxial split Hopkinson pressure bar device. The paper is well structured and can be accepted for publication. Some typos should be coorected:

Heading 2: “ANANLYSIS” should be “ANALYSIS”

Line 53 on page 11: It should be “bars are shown in FIG. 13.”

Line 63 in page 11: It should read: “FIG. 13 also shows the wave profiles”

Author Response

Thank you very much for your positive comments on this manuscript, and we have also revised the manuscript according to your suggestions. Details can be seen as follows:

Question 1:

“Heading 2: “ANANLYSIS” should be “ANALYSIS”.”

Response 1:

Thank you for this comment. According to your advice, we have changed “ANANLYSIS” to “ANALYSIS”. Details can be seen in this revised manuscript.

Question 2:

“Line 53 on page 11: It should be “bars are shown in FIG. 13.”

Response 2:

Thank you for this comment. According to your advice, we have made corresponding changes. Details can be seen in this revised manuscript.

Question 3:

“Line 63 in page 11: It should read: “FIG. 13 also shows the wave profiles””

Response 3:

Thank you for this comment. According to your advice, we have made corresponding changes. Details can be seen in this revised manuscript.

Reviewer 3 Report

In this paper, a wedge-shaped dual-wave bar (DWB) was designed and a biaxial SHPB experimental device was developed to solve the problem of propagation coupling between shear and axial stress waves.

The paper is well written and the thread is clear.The experimental testing is very interesting.

The paper is of sufficient novelty, even if  an important phenomenon such as the dynamic fracture was not treated. 

However, the introduction should be more coincise. The title of the heading 4 is very vague. Besides, refer the equations to well known citations.

Figs.3,4,5,6,7,8,9,13,14,15 aren't clearly visible and must be improved.  

Author Response

Thank you very much for your positive comments on this manuscript, and we have also revised the manuscript according to your suggestions. Details can be seen as follows:

Question 1:

“Moderate English changes required”

Response 1:

Thank you for this comment. According to your advice, English expression has been carefully improved throughout the manuscript. Details can be seen in this revised manuscript.

Question 2:

“However, the introduction should be more coincise. The title of the heading 4 is very vague. Besides, refer the equations to well known citations.”

Response 2:

Thank you for this comment. According to your advice, we have changed the title of the heading 4,  “4. THEORY” to “4. THEORY STUDY OF BIAXIAL IMPACT LOADING”, and cited related literatures as the description of the equations. Details can be seen in this revised manuscript.

Question 3:

“Figs.3,4,5,6,7,8,9,13,14,15 aren't clearly visible and must be improved.”

Response 3:

Thank you for this comment. According to your advice, we have reworked the high definition images and replaced the unclear ones in the manuscript. Details can be seen in this revised manuscript.

Round 2

Reviewer 1 Report

The authors have incorporated the suggestions very nicely. The manuscript may be accepted in the present form.

Reviewer 3 Report

The paper has been sufficiently revised .I think that the paper deserves the publication .